# INFERENCE, FAST AND SLOW: REINTERPRETING VAEs FOR OOD DETECTION

## ABSTRACT

Although likelihood-based methods are theoretically appealing, deep generative models (DGMs) often produce unreliable likelihood estimates in practice, particularly for out-of-distribution (OOD) detection. We reinterpret variational autoencoders (VAEs) through the lens of *fast and slow weights*. Our approach is guided by the proposed *Likelihood Path (LPath) Principle*, which extends the classical likelihood principle. A critical decision in our method is the selection of statistics for classical density estimation algorithms. The sweet spot should contain just enough information that's sufficient for OOD detection but not too much to suffer from the curse of dimensionality. Our LPath principle achieves this by selecting the sufficient statistics that form the "path" toward the likelihood. We demonstrate that this likelihood path leads to SOTA OOD detection performance, even when the likelihood itself is unreliable.

## 1 INTRODUCTION

Independent and identically distributed (IID) samples during training and testing are key to much of machine learning's (ML) success. However, as ML systems are deployed in the real world, encountering out-of-distribution (OOD) data is inevitable and poses significant safety risks. This is particularly challenging in the most general setting where labels are absent, and test input arrives in a streaming fashion. The objective of general *unsupervised OOD detection* is to develop a scalar score function, trained on $P_{\text{ID}}$ (in-distribution (ID) samples), that assigns higher scores to data from $P_{\text{OOD}}$ (out-of-distribution samples) than to data from $P_{\text{ID}}$.

Naïve approaches, such as using $p_\theta(\mathbf{x})$, the likelihood of deep generative models (DGMs), are attractive in theory but have proven ineffective due to unreliable likelihood estimates, often assigning high likelihood to OOD data (Nalisnick et al., 2018). Furthermore, even with perfect density estimation, likelihood alone is insufficient to detect OOD data (Le Lan & Dinh, 2021; Zhang et al., 2021) when the ID and OOD distributions overlap. Compounding this, recent theoretical works (Behrmann et al., 2021; Dai et al.) show that perfect density estimation may be infeasible for many DGMs.

> Research Question (RQ) 1: Can we achieve state-of-the-art (SOTA) unsupervised OOD detection without relying on accurate likelihood estimation?

We take a step towards answering this question by developing a *principled* method for unsupervised OOD detection. Our algorithm is inspired by a reinterpretation of Variational Autoencoders (VAEs) from the *fast and slow weights perspective*, originally proposed in the context of adaptive neural networks and meta-learning (Hinton & Plaut, 1987; Munkhdalai & Trischler, 2018; Ba et al., 2016). Our algorithm has two stages. In the first stage (**neural feature extraction**), we train VAEs and extract key statistics contributing to the likelihood function. In the second stage (**classical density estimation**), these statistics are used as training data to fit a classical statistical density estimation algorithm (COPOD (Li et al., 2020) or MD (Lee et al., 2018; Maciejewski et al., 2022)) for OOD detection.

The key design decision in our algorithm is the choice of statistics, which leads to our second research question:

> RQ 2: How do we select key statistics for the classical density estimation algorithm?

The desired statistics should strike a balance: including too many activations leads to the curse of dimensionality, while including too few fails to capture enough information. Our approach is to select the *minimal sufficient* statistics of the main components on the computational graph leading to the likelihood function. These anchoring statistics define the computational path of the likelihood function, which we term the *Likelihood Path (LPath) Principle*.

> Under imperfect likelihood estimation, there is more information in the computational path leading to the marginal likelihood function $p_\theta(\mathbf{x})$. Information can be optimally extracted by the *minimal sufficient statistics* of the individual components of the factorization of the likelihood function.

Although the LPath principle has independent interest in representation learning and can be applied to other DGMs, this work focuses on a thorough case study of applying the LPath principle to the OOD detection problem using Gaussian VAEs. We take the sufficient statistics of the VAE encoder and decoder as key statistics for our two-stage algorithm, achieving SOTA performance on common benchmarks (Table 1). Compared to other SOTA methods, we used a much smaller model (DC-VAEs from Xiao et al. (2020)'s architecture) with a parameter count of **3M**, compared to **44M** for Glow in DoSE (Morningstar et al., 2021) and **46M** for the diffusion model (Liu et al., 2023). We believe this "achieving more with less" phenomenon demonstrates our method's potential.

To summarize, our main contributions are:

**Empirical contribution:** We achieved SOTA unsupervised OOD detection performance on common benchmarks (Table 1) using a much smaller model compared to other SOTA methods, addressing RQ1.

**Methodological contribution:** We proposed the LPath Principle, which generalizes the classical likelihood principle[1] for instance-dependent inference (e.g., OOD detection) under imperfect density estimation, addressing RQ2.

## 2 INFERENCE, FAST AND SLOW

In this section, we reinterpret VAEs from the perspective of fast and slow weights. We begin by clearly distinguishing between likelihood evaluation and parameter inference procedures, as this distinction will be important throughout the paper.

**Inferential Procedure**  Given training data $\mathbf{X}_{\text{Train}} = \{\mathbf{x}_i\}_{i=1}^N$ and a density model $p_{\text{Model}} = p_\psi$ parameterized by $\psi$, we train $p_\psi$ on $\mathbf{X}_{\text{Train}}$ to obtain $p_{\psi_{\text{trained}}}$. This is an *inferential procedure*, transferring knowledge from $\mathbf{X}_{\text{Train}}$ to the trained parameters $\psi_{\text{trained}}$:

$$(\mathbf{X}_{\text{Train}}, p_\psi) \longrightarrow \psi_{\text{trained}} \in \Psi, \tag{1}$$

where $\Psi$ is the parameter space.

**Evaluation Procedure**  Suppose we have a new sample $\mathbf{x}$; we can compute the likelihood of $\mathbf{x}$ under the trained model $p_\psi$. This is an *evaluation procedure*, assessing $\mathbf{x}$ using the knowledge gained from training:

$$(\mathbf{x}, \psi_{\text{trained}}) \longrightarrow p_{\psi_{\text{trained}}}(\mathbf{x}) \in \mathbb{R}. \tag{2}$$

This typically occurs during test-time likelihood evaluation, after training is completed. However, direct application of this likelihood evaluation can assign higher likelihoods to OOD data than to ID data (Nalisnick et al., 2018).

While the evaluation procedure returns a scalar, the inferential procedure outputs a density model or parameters that characterize a model.

### 2.1 VAEs BACKGROUND

We next concrete examples of conditional distributions parameterized by encoder $q_\phi(\mathbf{z} \mid \mathbf{x})$ and decoder $p_\theta(\mathbf{x} \mid \mathbf{z})$ neural networks, as well as the prior. We choose Gaussian VAEs for illustration

---

[1]The marginal likelihood $p_\theta(\mathbf{x})$ is a special case, as it only uses the endpoint in the likelihood path.

because they are widely used and have very simple *minimal sufficient statistics*. If the reader is unfamilair with VAEs, see a more basic refresher of VAEs in Appendix A.

In our setup, the prior distribution is a standard Gaussian distribution:

$$p(\mathbf{z}) = \mathcal{N}(\mathbf{z} \mid \boldsymbol{\mu} = \mathbf{0}, \boldsymbol{\Sigma} = \mathbf{I}). \tag{3}$$

The encoder is a Gaussian distribution parameterized by an encoder neural network with parameters $\boldsymbol{\phi}$:

$$(\boldsymbol{\mu}_{\mathbf{z}}(\mathbf{x}), \boldsymbol{\sigma}_{\mathbf{z}}(\mathbf{x})) = \text{EncoderNeuralNet}_{\boldsymbol{\phi}}(\mathbf{x}), \tag{4}$$

$$q_{\boldsymbol{\phi}}(\mathbf{z} \mid \mathbf{x}) = \mathcal{N}\left(\mathbf{z} \mid \boldsymbol{\mu}_{\mathbf{z}}(\mathbf{x}), \text{diag}\left(\boldsymbol{\sigma}_{\mathbf{z}}^2(\mathbf{x})\right)\right). \tag{5}$$

Here, $(\boldsymbol{\mu}_{\mathbf{z}}(\mathbf{x}), \boldsymbol{\sigma}_{\mathbf{z}}(\mathbf{x}))$ are the *instance-dependent latent parameters* for the latent code $\mathbf{z}$. This inference occurs for every sample $\mathbf{x}$ and is the key property we aim to exploit.

The decoder is also a Gaussian distribution parameterized by a decoder neural network with parameters $\boldsymbol{\theta}$:

$$(\boldsymbol{\mu}_{\mathbf{x}}(\mathbf{z}), \boldsymbol{\sigma}_{\mathbf{x}}(\mathbf{z})) = \text{DecoderNeuralNet}_{\boldsymbol{\theta}}(\mathbf{z}), \tag{6}$$

$$p_{\boldsymbol{\theta}}(\mathbf{x} \mid \mathbf{z}) = \mathcal{N}\left(\mathbf{x} \mid \boldsymbol{\mu}_{\mathbf{x}}(\mathbf{z}), \text{diag}\left(\boldsymbol{\sigma}_{\mathbf{x}}^2(\mathbf{z})\right)\right). \tag{7}$$

Here, $\mathbf{z}$ is sampled from the encoder distribution $q_{\boldsymbol{\phi}}(\mathbf{z} \mid \mathbf{x})$. The pair $(\boldsymbol{\mu}_{\mathbf{x}}(\mathbf{z}), \boldsymbol{\sigma}_{\mathbf{x}}(\mathbf{z}))$ represents the *instance-dependent observable parameters* for reconstructing the observation $\mathbf{x}$. The reconstruction error is given by $\|\mathbf{x} - \boldsymbol{\mu}_{\mathbf{x}}(\mathbf{z})\|$, measuring the difference between the original input and its reconstruction.

## 2.2 VAE Reinterpreted: The Fast and Slow Weights Perspective

Consider Gaussian VAE learning. Given training data $\mathbf{X}_{\text{Train}} = \{\mathbf{x}_i\}_{i=1}^N$, we train an encoder $q_{\boldsymbol{\phi}}(\mathbf{z} \mid \mathbf{x})$ and a decoder $p_{\boldsymbol{\theta}}(\mathbf{x} \mid \mathbf{z})$:

$$q_{\boldsymbol{\phi}}(\mathbf{z} \mid \mathbf{x}) = \mathcal{N}\left(\mathbf{z} \mid \boldsymbol{\mu}_{\mathbf{z}}(\mathbf{x}; \boldsymbol{\phi}), \text{diag}\left(\boldsymbol{\sigma}_{\mathbf{z}}^2(\mathbf{x}; \boldsymbol{\phi})\right)\right), \tag{8}$$

$$p_{\boldsymbol{\theta}}(\mathbf{x} \mid \mathbf{z}) = \mathcal{N}\left(\mathbf{x} \mid \boldsymbol{\mu}_{\mathbf{x}}(\mathbf{z}; \boldsymbol{\theta}), \text{diag}\left(\boldsymbol{\sigma}_{\mathbf{x}}^2(\mathbf{z}; \boldsymbol{\theta})\right)\right). \tag{9}$$

After training, the knowledge in $\mathbf{X}_{\text{Train}}$ is transferred to $\boldsymbol{\phi}_{\text{trained}} = \boldsymbol{\phi}(\mathbf{X}_{\text{Train}})$ and $\boldsymbol{\theta}_{\text{trained}} = \boldsymbol{\theta}(\mathbf{X}_{\text{Train}})$. This is the first inferential procedure:

$$(\mathbf{X}_{\text{Train}}, q_{\boldsymbol{\phi}}, p_{\boldsymbol{\theta}}) \longrightarrow (\boldsymbol{\phi}_{\text{trained}}, \boldsymbol{\theta}_{\text{trained}}) \in (\Phi, \Theta). \tag{10}$$

At test time, when a new observation $\mathbf{x}_{\text{Test}}$ is given, the encoder and decoder Gaussian parameters are inferred depending on $\mathbf{x}_{\text{Test}}$. This is the second inferential procedure:

$$(\mathbf{x}_{\text{Test}}, \boldsymbol{\phi}_{\text{trained}}, \boldsymbol{\theta}_{\text{trained}}) \longrightarrow (\boldsymbol{\mu}_{\mathbf{z}}(\mathbf{x}_{\text{Test}}; \boldsymbol{\phi}_{\text{trained}}), \boldsymbol{\sigma}_{\mathbf{z}}(\mathbf{x}_{\text{Test}}; \boldsymbol{\phi}_{\text{trained}}), \boldsymbol{\mu}_{\mathbf{x}}(\mathbf{z}_{\text{Test}}; \boldsymbol{\theta}_{\text{trained}}), \boldsymbol{\sigma}_{\mathbf{x}}(\mathbf{z}_{\text{Test}}; \boldsymbol{\theta}_{\text{trained}})). \tag{11}$$

There are two kinds of parameters involved. The parameters $\boldsymbol{\phi}_{\text{trained}}$ and $\boldsymbol{\theta}_{\text{trained}}$ do not change after training—they are the *slow weights*. The quantities $\boldsymbol{\mu}_{\mathbf{z}}(\mathbf{x}_{\text{Test}}; \boldsymbol{\phi}_{\text{trained}}), \boldsymbol{\sigma}_{\mathbf{z}}(\mathbf{x}_{\text{Test}}; \boldsymbol{\phi}_{\text{trained}})$, $\boldsymbol{\mu}_{\mathbf{x}}(\mathbf{z}_{\text{Test}}; \boldsymbol{\theta}_{\text{trained}}), \boldsymbol{\sigma}_{\mathbf{x}}(\mathbf{z}_{\text{Test}}; \boldsymbol{\theta}_{\text{trained}})$ are instance-dependent and are considered the *fast weights* (Hinton & Plaut, 1987; Ba et al., 2016). From this perspective, the second inferential procedure uses knowledge both from $\mathbf{X}_{\text{Train}}$ (slow weights) and the test-time instance $\mathbf{x}_{\text{Test}}$ (fast weights).

In the next section, we detail how to use these fast weights $T(\mathbf{x}, \mathbf{z}) = (\boldsymbol{\mu}_{\mathbf{x}}(\mathbf{z}), \boldsymbol{\sigma}_{\mathbf{x}}(\mathbf{z}), \boldsymbol{\mu}_{\mathbf{z}}(\mathbf{x}), \boldsymbol{\sigma}_{\mathbf{z}}(\mathbf{x}))$ for OOD detection.

## 3 OOD Detection with Fast and Slow Weights

In this section, we reinterpret a classical prior OOD detection method from the slow weight perspective and introduce our method from the fast weight perspective. We then detail our algorithm. In the next section, we provide a thorough analysis of our method's statistical and combinatorial foundations.

### 3.1 OOD Detection with VAE Slow Weights

**Reinterpreting the Likelihood Regret Method** The likelihood regret method for OOD detection (Xiao et al., 2020) can be reinterpreted as detecting OOD samples using the information update in slow weights. At a high level, after obtaining $\theta_{\text{trained}}$ from training, they fine-tune VAEs by maximizing likelihood on a test sample $\mathbf{x}_{\text{Test}}$ to get $\theta_{\text{online}}$, and track the following likelihood regret:

$$\log p(\theta_{\text{online}} \mid \mathbf{x}_{\text{Test}}) - \log p(\theta_{\text{trained}} \mid \mathbf{x}_{\text{Test}}). \tag{12}$$

In other words, their work involves two inferential procedures. First, $(\mathbf{X}_{\text{Train}}, p_\theta) \longrightarrow \theta_{\text{trained}}$; second, $(\mathbf{X}_{\text{Train}}, \mathbf{x}_{\text{Test}}, p_\theta) \longrightarrow \theta_{\text{online}}$, where they do not maximize $p_\theta$ jointly on $(\mathbf{X}_{\text{Train}}, \mathbf{x}_{\text{Test}})$, but sequentially on $\mathbf{X}_{\text{Train}}$ first and $\mathbf{x}_{\text{Test}}$ next. However, likelihood regret is empirically outperformed by alternative approaches (Morningstar et al., 2021) which did not involve any fine-tuning. This is probably because training neural networks on one sample is challenging. Optimizing for a few iterations changes $\theta_{\text{trained}}$ very little, while training for many iterations results in overfitting quickly. Furthermore, in streaming OOD detection, such computational overhead is formidable.

### 3.2 OOD Detection with VAE Fast Weights

Given that OOD detection with slow weights induces formidable computational overhead during test time and poses optimization challenges, we propose to perform OOD detection with fast weights. In Section 2, we reinterpreted the encoder and decoder means and variances as the fast weights of the VAE: $T(\mathbf{x}, \mathbf{z}) = (\boldsymbol{\mu}_{\mathbf{x}}(\mathbf{z}), \boldsymbol{\sigma}_{\mathbf{x}}(\mathbf{z}), \boldsymbol{\mu}_{\mathbf{z}}(\mathbf{x}), \boldsymbol{\sigma}_{\mathbf{z}}(\mathbf{x}))$. However, these remain high-dimensional. This not only increases computational time but can also cause issues for the second-stage statistical algorithm (Maciejewski et al., 2022). We address this problem by taking the L2 norm of $T(\mathbf{x}, \mathbf{z})$:

$$u(\mathbf{x}) = \|\mathbf{x} - \widehat{\mathbf{x}}\|_2 = \|\mathbf{x} - \boldsymbol{\mu}_{\mathbf{x}}(\boldsymbol{\mu}_{\mathbf{z}}(\mathbf{x}))\|_2, \tag{13}$$

$$v(\mathbf{x}) = \|\boldsymbol{\mu}_{\mathbf{z}}(\mathbf{x})\|_2, \tag{14}$$

$$w(\mathbf{x}) = \|\boldsymbol{\sigma}_{\mathbf{z}}(\mathbf{x})\|_2, \tag{15}$$

$$s(\mathbf{x}) = \|\boldsymbol{\sigma}_{\mathbf{x}}(\boldsymbol{\mu}_{\mathbf{z}}(\mathbf{x}))\|_2. \tag{16}$$

Note that in Eq. 13, instead of taking $\|\boldsymbol{\mu}_{\mathbf{x}}(\boldsymbol{\mu}_{\mathbf{z}}(\mathbf{x}))\|_2$, we compute $\|\mathbf{x} - \boldsymbol{\mu}_{\mathbf{x}}(\boldsymbol{\mu}_{\mathbf{z}}(\mathbf{x}))\|_2$. This is because $\|\boldsymbol{\mu}_{\mathbf{x}}(\boldsymbol{\mu}_{\mathbf{z}}(\mathbf{x}))\|_2$ could be unnormalized in magnitude compared to other statistics, causing problems in the second-stage classical density estimation algorithm. Thus, we normalize it by taking the reconstruction error, which should be close to zero due to the VAE optimization objective. While VAE optimization should already be driving Eqs. 14–16 to a small value.

### 3.3 The LPath Algorithm for Fast Weights OOD Detection

We use Eqs. 13–16 as the scoring metrics for our OOD detection algorithm. We call it the Likelihood Path (LPath) algorithm because it is based on minimal sufficient statistics of the individual components of the factorization of the likelihood function; we provide a detailed description and analysis in Section 4.3.

Our algorithm is detailed in Algorithm 1. It first trains a VAE and extracts statistics in Eqs. 13–16 in the first stage (**neural feature extraction**). Then it fits a classical statistical density estimation algorithm (COPOD (Li et al., 2020) or MD (Lee et al., 2018; Maciejewski et al., 2022)) in the second stage (**classical density estimation**).

Our algorithm can be used with a single VAE model (LPath-1M) or a pair of two models (LPath-2M). For LPath-1M, we use the same VAE to extract all of $u(\mathbf{x}), v(\mathbf{x}), w(\mathbf{x}), s(\mathbf{x})$. When used with a pair of two models (LPath-2M), we train two VAEs: one with a very high latent dimension (e.g., 1000) and another with a very low dimension (e.g., 1 or 2). In the second stage, we extract the following statistics: $(u(\mathbf{x})_{\text{lowD}}, v(\mathbf{x})_{\text{highD}}, w(\mathbf{x})_{\text{highD}}, s(\mathbf{x})_{\text{lowD}})$, where $u(\mathbf{x})_{\text{lowD}}, s(\mathbf{x})_{\text{lowD}}$ are taken from the low-dimensional VAE and $v(\mathbf{x})_{\text{high D}}, w(\mathbf{x})_{\text{high D}}$ from the high-dimensional VAE. Appendix D.1.2 explains the reasoning behind this combination.

---

**Algorithm 1** Training and Inference: From high-dimensional data to OOD scoring

---

1: **Input:** $D_{\text{Train, ID}} \sim P_{\text{ID}}$ of size $n_{\text{train}} \times n_{\text{channels}}$, $D_{\text{Test}} \sim P_{\text{ID}} \cup P_{\text{OOD}}$
2: **Stage 1 Training: From high-dim dataset to low-dim minimal sufficient statistics**
3: Train VAE on $D_{\text{Train, ID}}$                                    ▷ Normal training using SGD/Adam
4: **for** $\mathbf{x} \in D_{\text{Train, ID}}$ **do**
5:     Compute and record $T(\mathbf{x}) = (u(\mathbf{x}), v(\mathbf{x}), w(\mathbf{x}), s(\mathbf{x}))$          ▷ As in Eqs. 13–15
6: **end for**
7: Create new dataset $D_{\text{Train, ID, T}}$ of size $n_{\text{train}} \times 4$ consisting of the minimal sufficient statistics $T(\mathbf{x})$
    for Stage 2 Training
8: **Stage 2 Training: From low-dim minimal sufficient statistics to OOD scoring**
9: Select classical OOD scoring algorithm $\mathcal{A}_{\text{Classical}}$ (e.g., COPOD (Li et al., 2020) or MD (Lee
    et al., 2018))
10: Train $\mathcal{A}_{\text{Classical}}$ on $D_{\text{Train, ID, T}}$ to get $\mathcal{A}_{\text{Classical, Trained}}$          ▷ Classical OOD training
11: **Inference Stage: OOD Scoring**
12: **for** $\mathbf{x}_{\text{Test}} \in D_{\text{Test}}$ **do**
13:     Compute $T(\mathbf{x}_{\text{Test}}) = (u(\mathbf{x}_{\text{Test}}), v(\mathbf{x}_{\text{Test}}), w(\mathbf{x}_{\text{Test}}), s(\mathbf{x}_{\text{Test}}))$ from trained VAE
14:     Compute OOD score $S(\mathbf{x}_{\text{Test}}) = \mathcal{A}_{\text{Classical, Trained}}(T(\mathbf{x}_{\text{Test}}))$
15: **end for**
16: **Output:** $S(\mathbf{x}_{\text{Test}})$, an OOD score for each $\mathbf{x}_{\text{Test}}$

---

# 4 THE LIKELIHOOD PATH PRINCIPLE

In this section, we provide an in-depth analysis of how we arrived at our selected $T(\mathbf{x}, \mathbf{z}) = (\boldsymbol{\mu}_{\mathbf{x}}(\mathbf{z}), \boldsymbol{\sigma}_{\mathbf{x}}(\mathbf{z}), \boldsymbol{\mu}_{\mathbf{z}}(\mathbf{x}), \boldsymbol{\sigma}_{\mathbf{z}}(\mathbf{x}))$, the fundamental challenge for this problem, and how to have a general principle to select such statistics not just for VAEs but for other DGMs.

Recall RQ2:

RQ 2: How do we select key statistics for the classical density estimation algorithm?

The goal is to overcome the challenge of dimensionality: If the dimensionality is too high, we might suffer from the curse of dimensionality, but if the dimensionality is too low, we might capture insufficient information to make effective inference. How do we find the sweet spot?

The key idea is our proposed Likelihood Path (LPath) Principle:

Under imperfect likelihood estimation, there is more information in the computational path leading to the marginal likelihood function $p_\theta(\mathbf{x})$. Information can be optimally extracted by the *minimal sufficient statistics* of the individual components of the factorization of the likelihood function.

The LPath Principle is a general-purpose principle to select such statistics, and our analysis could also be used to select such statistics for other DGMs; we leave that for future work.

We will start by reviewing the Likelihood Principle and Sufficiency Principle that form the statistical foundation of our proposed LPath Principle.

## 4.1 THE LIKELIHOOD PRINCIPLE

The maximum likelihood estimation (MLE) approach to unsupervised learning focuses on finding parameters $\psi$ to maximize the likelihood $\ell(\psi \mid \mathbf{x}) := p(\mathbf{x} \mid \psi)$ given training data $\mathbf{x}$, so that we transfer information from $\mathbf{x}$ to $\psi$. MLE is a special case of the *likelihood principle* (Berger & Wolpert, 1988):

The *likelihood principle* states that all the evidence in an observed sample $\mathbf{x}$ relevant to model parameters is contained in the likelihood function $\ell(\psi \mid \mathbf{x})$.

MLE satisfies the likelihood principle because inferring the most likely parameter depends only on $\ell(\psi \mid \mathbf{x})$. Many OOD detection works (Nalisnick et al., 2019; Xiao et al., 2020) satisfy this principle as well.

*In summary, the likelihood principle postulates that $\ell(\psi \mid \mathbf{x})$ (as a function of $\psi$) tells us everything about $\mathbf{x}$. If we make our decisions (e.g., OOD detection) based only on the likelihood function, our decision satisfies the likelihood principle.*

## 4.2 THE SUFFICIENCY PRINCIPLE

While the likelihood principle suggests that all information is contained in $\ell(\psi \mid \mathbf{x})$, it is a complex function and does not directly tell us what statistics to include for RQ2. To better process such overwhelming information, we seek to reduce our selection to the simplest set that still contains sufficient information about $\ell(\psi \mid \mathbf{x})$. How do we formalize such information trimming in the context of unsupervised learning?

- The information reduction procedure $T$ should be a function of $\mathbf{x}$, a *statistic*.
- $T$ should be *sufficient* for describing $p(\mathbf{x} \mid \psi)$ or $\psi$: $p(\mathbf{x} \mid T(\mathbf{x}), \psi) = p(\mathbf{x} \mid T(\mathbf{x}))$.
- $T$ should be *minimal*: $F(T)$ is no longer sufficient for $\psi$, for any non-invertible function $F$.

*In summary, a minimal sufficient statistic $T$ tells us everything about $\psi$ that we can possibly learn from observing $\mathbf{x}$, and if we attempt to trim $T$ further by any irreversible process, we would lose some information for inferring $\ell(\psi \mid \mathbf{x})$ [2].*

Alternatively, we can view sufficient statistics from an information-theoretic perspective. Let I denote the mutual information. $T(\mathbf{x})$ is sufficient for $\psi$ if:

$$\mathrm{I}(\psi; T(\mathbf{x})) = \mathrm{I}(\psi; \mathbf{x}). \tag{17}$$

In other words, the data processing inequality $\mathrm{I}(\psi; T(\mathbf{x})) \leq \mathrm{I}(\psi; \mathbf{x})$ becomes an equality if $T$ is sufficient. This is useful for answering RQ2. Given a new sample $\mathbf{x}$, the encoder and decoder neural nets would produce millions of activations, all of which could be useful for OOD detection. However, this is clearly overwhelming. The minimal sufficient statistic $T(\mathbf{x})$ gives us the set of statistics that cannot be reduced further without losing some information.

The standard Gaussian VAE's encoder and decoder parameterizations by sample mean vectors and sample covariance matrices (Eqs. 4 and 6) are *minimal sufficient statistics* (Wasserman, 2006). Here, minimal sufficient statistics represent two **optimal** conditions for inference: They are *sufficient* because once $(\boldsymbol{\mu}_\mathbf{z}(\mathbf{x}), \boldsymbol{\sigma}_\mathbf{z}(\mathbf{x}))$ and $(\boldsymbol{\mu}_\mathbf{x}(\mathbf{z}), \boldsymbol{\sigma}_\mathbf{x}(\mathbf{z}))$ are known, the conditional likelihood functions can be defined. They are *minimal* because any other parameterization of a Gaussian will involve no fewer parameters.

## 4.3 LIKELIHOOD PATH PRINCIPLE

Our proposed LPath principle states that:

Under imperfect likelihood estimation, there is more information in the computational path leading to the marginal likelihood function $p_\theta(\mathbf{x})$. Information can be optimally extracted by the *minimal sufficient statistics* of the individual components of the factorization of the likelihood function.

For VAEs, this entails applying the *likelihood principle* twice in the VAE's encoder and decoder conditional distributions and tracking their *minimal sufficient statistics*: $T(\mathbf{x}, \mathbf{z}) = (\boldsymbol{\mu}_\mathbf{x}(\mathbf{z}), \boldsymbol{\sigma}_\mathbf{x}(\mathbf{z}), \boldsymbol{\mu}_\mathbf{z}(\mathbf{x}), \boldsymbol{\sigma}_\mathbf{z}(\mathbf{x}))$.

Recall the VAE formulation:

$$\boxed{\text{LHS has no closed form likelihood nor sufficient statistics.}} \quad \log p_\theta(\mathbf{x}) \approx \log \left[ \frac{p_\theta(\mathbf{x} \mid \mathbf{z}) \, p(\mathbf{z})}{q_\phi(\mathbf{z} \mid \mathbf{x})} \right] \quad \boxed{\text{RHS contains more information given by their minimal sufficient statistics.}} \tag{18}$$

While it is not obvious how to apply likelihood and sufficiency principles to the VAE's marginal likelihood $p_\theta(\mathbf{x})$, we can apply them to the Gaussian VAE's encoder $q_\phi(\mathbf{z} \mid \mathbf{x})$, prior $p(\mathbf{z})$, and decoder $p_\theta(\mathbf{x} \mid \mathbf{z})$, which completely characterize $p_\theta(\mathbf{x})$.

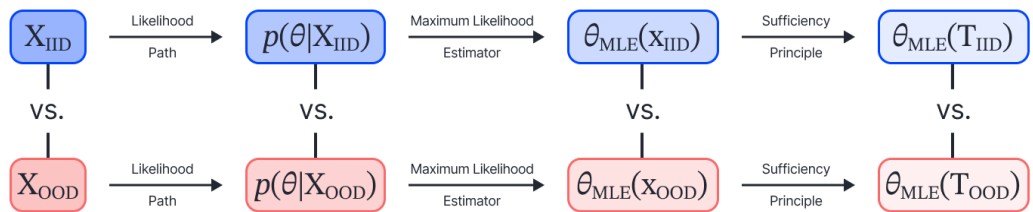

Figure 1: Left to right shows the information reduction via the likelihood principle (LP), maximum likelihood estimation (MLE), and sufficiency principle (SP). $T$ denotes sufficient statistics. The top and bottom rows contrast inferences between $\mathbf{x}_{\text{ID}}$ and $\mathbf{x}_{\text{OOD}}$.

Let us make the above precise in our VAE's LPath. Consider the following Markov chain when we estimate the marginal likelihood of a sample $\mathbf{x}$:

$$\mathbf{x} \longrightarrow q_\phi(\mathbf{z} \mid \mathbf{x}), p(\mathbf{z}), p_\theta(\mathbf{x} \mid \mathbf{z}) \longrightarrow p_\theta(\mathbf{x}). \tag{19}$$

The data processing inequality from information theory says:

$$\text{I}(\mathbf{x}; (q_\phi(\mathbf{z} \mid \mathbf{x}), p(\mathbf{z}), p_\theta(\mathbf{x} \mid \mathbf{z}))) \geq \text{I}(\mathbf{x}; p_\theta(\mathbf{x})). \tag{20}$$

When density estimation is perfect, the above inequality becomes an equality. In practical cases, perfect learning never happens. Mathematically, our LPath principle thus states:

$$\text{I}(\mathbf{x}; (q_\phi(\mathbf{z} \mid \mathbf{x}), p(\mathbf{z}), p_\theta(\mathbf{x} \mid \mathbf{z}))) > \text{I}(\mathbf{x}; p_\theta(\mathbf{x})). \tag{21}$$

In a nutshell, ***the central theme in our work is to exploit the gap in Inequality 21.***

The chain of information reduction for OOD inference and detection is summarized by Figure 1:

In the first column of Figure 1, it is hard to define a metric in the visible space to distinguish $\mathbf{x}_{\text{ID}}$ and $\mathbf{x}_{\text{OOD}}$, even though they contain the most evidence. In the second column, we compare them by comparing their corresponding likelihood functions, suggested by the likelihood principle. The third column compares their maximum likelihood inferences. The last column suggests that it suffices to know the sufficient statistics $T$ to obtain $\theta_{\text{MLE}}$, which completes the information reduction chain.

### 4.4 COMBINATORIAL CANCELLATION

We analyzed the LPath Principle for OOD detection from the statistical perspective. We can gain more concrete insights on why the LPath Principle works if we take a combinatorial perspective, which can act as an empirical method to select statistics, answering RQ2. The key insight is that factors in the likelihood function risk **getting canceled** in the likelihood itself, and the signals they contain for OOD detection will be drowned out. This is how information is lost in Eq. 21. To address this, we could separate each factor out and capture the signal they contain with their sufficient statistics, arriving at our LPath Principle.

In the case of VAEs, the encoder and decoder contain complementary information for OOD detection, but they could be canceled out in $\log p_\theta(\mathbf{x})$. Recall the VAE's likelihood estimation:

$$\log p_\theta(\mathbf{x}) \approx \log \left[ \frac{p_\theta(\mathbf{x} \mid \mathbf{z}) \, p(\mathbf{z})}{q_\phi(\mathbf{z} \mid \mathbf{x})} \right].$$

The decoder's conditional likelihood $p_\theta(\mathbf{x} \mid \mathbf{z})$ being too large and prior $p(\mathbf{z})$ (evaluated at samples from the encoder $q_\phi(\mathbf{z} \mid \mathbf{x})$) being too small both suggest $\mathbf{x}$ could be an anomaly, but their scalar product can be well-ranged, which drowns out the signal for OOD discovery. A more concrete interpretation of this cancellation phenomenon from the pixel texture vs. semantics perspective can be found in Appendix C.

For $\mathbf{x}_{\text{ID}}$ and $\mathbf{x}_{\text{OOD}}$, we would anticipate different likelihood paths. This difference can be detected by their corresponding sufficient statistics: $T(\mathbf{x}_{\text{ID}}, \mathbf{z}_{\text{ID}}) = (\boldsymbol{\mu}_\mathbf{x}(\mathbf{z}_{\text{ID}}), \boldsymbol{\sigma}_\mathbf{x}(\mathbf{z}_{\text{ID}}), \boldsymbol{\mu}_\mathbf{z}(\mathbf{x}_{\text{ID}}), \boldsymbol{\sigma}_\mathbf{z}(\mathbf{x}_{\text{ID}}))$ and $T(\mathbf{x}_{\text{OOD}}, \mathbf{z}_{\text{OOD}}) = (\boldsymbol{\mu}_\mathbf{x}(\mathbf{z}_{\text{OOD}}), \boldsymbol{\sigma}_\mathbf{x}(\mathbf{z}_{\text{OOD}}), \boldsymbol{\mu}_\mathbf{z}(\mathbf{x}_{\text{OOD}}), \boldsymbol{\sigma}_\mathbf{z}(\mathbf{x}_{\text{OOD}}))$. In other words, a new sample may be considered as ID if its sufficient statistics are similar to $T(\mathbf{x}_{\text{ID}}, \mathbf{z}_{\text{ID}})$ for some $\mathbf{x}_{\text{ID}} \in P_{\text{ID}}$ (because the encoder and decoder distributions are completely characterized by $T$).

| ID
OOD | CIFAR10 | | | | SVHN | | | FMNIST | | | MNIST | | |
|---|---|---|---|---|---|---|---|---|---|---|---|---|---|
| | SVHN | CIFAR100 | Hflip | Vflip | CIAFR10 | Hflip | Vflip | MNIST | Hflip | Vflip | FMNIST | Hflip | Vflip |
| ELBO | 0.08 | 0.54 | 0.5 | 0.56 | **0.99** | 0.5 | 0.5 | 0.87 | 0.63 | 0.83 | **1.00** | 0.59 | 0.6 |
| LR (Xiao et al., 2020) | 0.88 | N/A | N/A | N/A | 0.92 | N/A | N/A | 0.99 | N/A | N/A | N/A | N/A | N/A |
| BIVA (Havtorn et al., 2021) | 0.89 | N/A | N/A | N/A | **0.99** | N/A | N/A | 0.98 | N/A | N/A | **1.00** | N/A | N/A |
| DoSE (Morningstar et al., 2021) | 0.97 | 0.57 | 0.51 | 0.53 | **0.99** | 0.52 | 0.51 | **1.00** | 0.66 | 0.75 | **1.00** | **0.81** | 0.83 |
| Fisher (Bergamin et al., 2022) | 0.87 | 0.59 | N/A | N/A | N/A | N/A | N/A | 0.96 | N/A | N/A | N/A | N/A | N/A |
| DDPM (Liu et al., 2023) | 0.98 | N/A | 0.51 | 0.63 | **0.99** | **0.62** | **0.58** | 0.97 | 0.65 | **0.89** | N/A | N/A | N/A |
| LMD (Graham et al., 2023) | **0.99** | 0.61 | N/A | N/A | 0.91 | N/A | N/A | 0.99 | N/A | N/A | **1.00** | N/A | N/A |
| LPath-1M-COPOD (Ours) | **0.99** | **0.62** | **0.53** | 0.61 | **0.99** | 0.55 | 0.56 | **1.00** | 0.65 | 0.81 | **1.00** | 0.65 | **0.87** |
| LPath-2M-COPOD (Ours) | 0.98 | **0.62** | **0.53** | **0.65** | 0.96 | 0.56 | 0.55 | 0.95 | **0.67** | 0.87 | **1.00** | 0.77 | 0.78 |
| LPath-1M-MD (Ours) | **0.99** | 0.58 | 0.52 | 0.60 | 0.95 | 0.52 | 0.52 | 0.97 | 0.63 | 0.82 | **1.00** | 0.75 | 0.76 |

Table 1: AUROC of OOD Detection with different ID and OOD datasets. LPath-1M is LPath with one model, LPath-2M is LPath with two models.

## 5 EXPERIMENTS

We compare our methods with state-of-the-art OOD detection methods (Kirichenko et al., 2020; Xiao et al., 2020; Havtorn et al., 2021; Morningstar et al., 2021; Bergamin et al., 2022; Liu et al., 2023; Graham et al., 2023), under the unsupervised, single batch, no data inductive bias assumption setting.

Following the convention in those methods, we have conducted experiments with a number of common benchmarks, including CIFAR10 (Krizhevsky & Hinton, 2009), SVHN (Netzer et al., 2011), CIFAR100 (Krizhevsky & Hinton, 2009), MNIST (LeCun et al., 1998), FashionMNIST (FMNIST) (Xiao et al., 2017), and their horizontally flipped and vertically flipped variants.

**Experimental Results.** Table 1 show that our methods surpass or are on par with the state-of-the-art (SOTA). Because our setting assumes no access to labels, batches of test data, or any inductive bias on the dataset, OOD datasets like Hflip and Vflip become very challenging. Most prior methods achieved only near-chance AUROC on Vflip and Hflip for CIFAR10 and SVHN as ID data. This is expected because horizontally flipped CIFAR10 or SVHN differs from the in-distribution only by one latent dimension. Even so, our methods still managed to surpass prior SOTA in some cases, though only marginally. More experimental details, including various ablation studies, are in Appendix D, E.

**Achieving More with Less.** This improvement is more significant given that we only used a very small VAE architecture. Compared to other SOTA methods, we used a much smaller model (DC-VAEs from (Xiao et al., 2020)'s architecture) with a parameter count of **3M**, compared to **44M** for Glow (Kingma & Dhariwal, 2018) in DoSE (Morningstar et al., 2021) and **46M** for the diffusion model (Rombach et al., 2022; Liu et al., 2023). Specifically, our method clearly exceeds other VAE-based methods (Xiao et al., 2020; Havtorn et al., 2021), and is the only VAE-based method that is competitive against bigger models. DoSE (Morningstar et al., 2021) conducted experiments on VAEs with five carefully chosen statistics. They reported their MNIST/FMNIST results on their VAEs and used Glow on more difficult datasets like CIFAR/SVHN. We assume the reason is that Glow performed better on more complex datasets. Our methods surpass their Glow-based results, which should, in turn, be better than their method applied to VAEs. On one hand, Glow's likelihood is arguably much better estimated than our small DC-VAE model. On the other hand, their statistics appear to be more sophisticated. However, our simple method manages to surpass their scores. This showcases the efficiency and effectiveness of our method.

## 6 CONCLUSION

We presented the Likelihood Path Principle applied to unsupervised, one-sample OOD detection. We provided in-depth analyses from the neural (fast-slow weights), statistical (likelihood and sufficiency principles), and combinatorial (cancellation effect) perspectives. Our method is principled and supported by SOTA results. In future work, we plan to adapt our principles and techniques to more powerful DGMs, such as Glow or diffusion models.

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

## A  VAEs Background

We use $P$ to denote distributions and $p$ as their associated densities. Variational Autoencoders (VAEs) (Kingma & Welling, 2013) are a distinct member of the family of deep generative models (DGMs), where the likelihood is computed by marginalizing the following joint model likelihood $p_\theta(\mathbf{x}, \mathbf{z})$, parameterized by $\theta$: $p_\theta(\mathbf{x}) = \int_{\mathbf{z} \sim P(\mathbf{z})} p_\theta(\mathbf{x}, \mathbf{z}) \, d\mathbf{z}$.

Here, $p_\theta(\mathbf{x})$ is called the marginal likelihood and is treated as a function of $\theta$. VAEs are classified as *latent variable models* (Kingma et al., 2019), where latent variables $\mathbf{z}$ represent unobserved random variables modeled as the source of the data-generating process. The marginal likelihood can be expressed as:

$$p_\theta(\mathbf{x}) = \int_{\mathbf{z} \sim P(\mathbf{z})} p_\theta(\mathbf{x}, \mathbf{z}) \, d\mathbf{z} = \int_{\mathbf{z} \sim P(\mathbf{z})} p_\theta(\mathbf{x} \mid \mathbf{z}) p(\mathbf{z}) \, d\mathbf{z}. \tag{22}$$

When both the prior $P(\mathbf{z})$ and the conditional distribution $P_\theta(\mathbf{x} \mid \mathbf{z})$ are Gaussian, the marginal likelihood $p_\theta(\mathbf{x})$ can be thought of as an infinite Gaussian mixture model, making it highly expressive. However, in high-dimensional settings (e.g., images), directly estimating $\log p_\theta(\mathbf{x}) = \log [p_\theta(\mathbf{x} \mid \mathbf{z}) p(\mathbf{z})] \approx \log(\frac{1}{K} \sum_{k=1}^{K} [p_\theta(\mathbf{x} \mid \mathbf{z}_k) p(\mathbf{z}_k)])$ with finite samples becomes computationally inefficient. VAEs introduce an efficient sampling method via an encoder $q_\phi(\mathbf{z} \mid \mathbf{x})$ that serves as an importance-weighted sampler, making computation much more tractable. This is formalized as:

$$p_\theta(\mathbf{x}) = \int_{\mathbf{z} \sim P(\mathbf{z})} p_\theta(\mathbf{x} \mid \mathbf{z}) p(\mathbf{z}) \, d\mathbf{z} = \int_{\mathbf{z} \sim q_\phi(\mathbf{z}|\mathbf{x})} \frac{p_\theta(\mathbf{x} \mid \mathbf{z}) p(\mathbf{z})}{q_\phi(\mathbf{z} \mid \mathbf{x})} \, d\mathbf{z}, \tag{23}$$

with a one-sample approximation:

$$\log p_\theta(\mathbf{x}) \approx \log \left[ \frac{p_\theta(\mathbf{x} \mid \mathbf{z}) p(\mathbf{z})}{q_\phi(\mathbf{z} \mid \mathbf{x})} \right]. \tag{24}$$

For out-of-distribution (OOD) detection, we utilize the test-time latent variable inference of VAEs, so we omit the training dynamics here. For more details on VAEs, see Doersch (2016); Kingma et al. (2019).

## B  Related Work

Prior works have approached OOD detection from various perspectives and with different data assumptions, e.g., with or without access to training labels, batches of test data, or single test data points in a streaming fashion, and with or without knowledge and inductive bias of the data. In the following, we give an overview organized by different data assumptions with a focus on where our method fits.

The first assumption is whether the method has access to training labels. There has been extensive work on classifier-based methods that assume access to training labels (Hendrycks & Gimpel, 2016; Frosst et al., 2019; Sastry & Oore, 2020; Bahri et al., 2021; Papernot & McDaniel, 2018; Osawa et al., 2019; Guénais et al., 2020; Lakshminarayanan et al., 2016; Pearce et al., 2020). Within this category, there are different assumptions as well, such as access to a pretrained network or knowledge of OOD test examples. See Table 1 of Sastry & Oore (2020) for a summary of such methods.

When we do not assume access to the training labels, the problem becomes more general and also harder. Under this category, some methods assume access to a batch of test data where either all the data points are OOD or not (Nalisnick et al., 2019). A more general setting does not assume OOD data would come in batches. Under this setup, there are methods that implicitly assume prior knowledge of the data, such as the input complexity method (Serrà et al., 2019), where the use of image compressors implicitly assumes an image-like structure, or the likelihood ratio method (Ren

et al., 2019), where a noisy background model is trained with the assumption of a background-object structure.

As mentioned in Section 1, our method is among the most general and difficult settings where we assume no access to labels, batches of test data, or any inductive bias of the dataset (Xiao et al., 2020; Kirichenko et al., 2020; Havtorn et al., 2021; Ahmadian & Lindsten, 2021; Morningstar et al., 2021; Bergamin et al., 2022; Liu et al., 2023; Graham et al., 2023). Xiao et al. (2020) fine-tune the VAE encoders on the test data and take the likelihood ratio as the OOD score. Kirichenko et al. (2020) trained RealNVP on EfficientNet (Tan & Le, 2020) embeddings and use log-likelihood directly as the OOD score. Havtorn et al. (2021) trained hierarchical VAEs such as HVAE and BIVA and used the log-likelihood directly as the OOD score. We compare our method with the above methods in Table 1.

Some recent works on OOD detection (Ahmadian & Lindsten, 2021; Bergamin et al., 2022; Morningstar et al., 2021; Graham et al., 2023; Liu et al., 2023; Osada et al., 2023) indeed start to consider other information contained in the entire neural activation path leading to the likelihood. Examples include entropy, KL divergence, and Jacobian in the likelihood (Morningstar et al., 2021). However, they do not address RQ2 and provide a principled method to select such statistics.

## C  INTERPRETATION OF LIKELIHOOD CANCELLATION

Recall VAEs' likelihood estimation (parameterized by $\theta$):

$$\log p_\theta(\mathbf{x}) \approx \log \left[ \frac{p_\theta(\mathbf{x} \mid \mathbf{z}) \, p(\mathbf{z})}{q_\phi(\mathbf{z} \mid \mathbf{x})} \right], \tag{25}$$

The decoder $p_\theta(\mathbf{x} \mid \mathbf{z})$'s reconstruction focuses on the pixel textures, while encoder $q_\phi(\mathbf{z} \mid \mathbf{x})$'s samples evaluated at the prior, $p(\mathbf{z})$, describe semantics. Consider $\mathbf{x}_{\text{OOD}}$, whose lower level features are similar to ID data, but is semantically different. We can imagine $p_\theta(\mathbf{x} \mid \mathbf{z})$ is large while $p(\mathbf{z})$ is small. However, (Havtorn et al., 2021) demonstrates $p_\theta(\mathbf{x})$ is dominated by lower level information. Even if $p(\mathbf{z})$ wants to reveal $\mathbf{x}_{\text{OOD}}$'s OOD nature, we cannot decipher it through $p_\theta(\mathbf{x}_{\text{OOD}})$. The converse: $p_\theta(\mathbf{x} \mid \mathbf{z})$ can flag $\mathbf{x}_{\text{OOD}}$ when the reconstruction error is big. But if $p(\mathbf{z})$ is unusually high compared to typical $\mathbf{x}_{\text{ID}}$, $p_\theta(\mathbf{x})$ may appear less OOD.

## D  EXPERIMENTAL DETAILS

### D.1  VAE ARCHITECTURE AND TRAINING

For the architecture and the training of our VAEs, we followed Xiao et al. (2020). In addition, we have trained VAEs of varying latent dimensions, {1, 2, 5, 10, 100, 1000, 2000, 3096, 5000, 10000}, and instead of training for 200 epochs and taking the resulting model checkpoint, we took the checkpoint that had the best validation loss. For LPath-1M, we conducted experiments on VAEs with all latent dimensions and for LPath-2M, we paired one high-dimensional VAE from the group {3096, 5000, 10000} and one low-dimensional VAE from the group {1, 2, 5}.

In addition to Gaussian VAEs as mentioned in Section D.1.3, we also empirically experimented with a categorical decoder, in the sense the decoder output is between the discrete pixel ranges, as in Xiao et al. (2020). Strictly speaking, this no longer satisfies the Gaussian distribution anymore, which may in turn violate our sufficient statistics perspective. However, we still experimented with it to test whether LPath principles can be interpreted as a heuristic to inspire methods that approximate sufficient statistics that can work reasonably well, and we observed that categorical decoders work similarly with Guassian decoders.

### D.1.1  DIMENSIONALITY TRADE-OFF

In this section, we discuss heuristics for training VAEs in the context of OOD detection, focusing on the trade-offs involved in selecting the latent dimension.

**Balancing the Trade-off in Latent Dimension**  A single VAE encounters a trade-off when selecting the latent dimension for effective OOD detection:

- **Higher Latent Dimension Benefits the Encoder**: Increasing the latent dimension enhances the encoder's ability $q_\phi$ to discriminate between in-distribution (ID) and OOD data. A higher-dimensional latent space allows the encoder to map ID and OOD data to more distinct regions, reducing overlap and improving separability. This increased capacity enables the encoder to capture complex features of the data, improving its discriminative power.

- **Lower Latent Dimension Benefits the Decoder**: Decreasing the latent dimension enhances the decoder's ability $p_\theta$ to identify OOD data through reconstruction errors. A lower-dimensional latent space constrains the decoder, making it less capable of accurately reconstructing OOD data that it hasn't seen during training. This constraint leads to larger reconstruction errors $u(\mathbf{x}) = \|\mathbf{x} - \hat{\mathbf{x}}\|_2$ for OOD samples, providing a useful signal for detection.

This trade-off poses a challenge: adjusting the latent dimension to favor one component (encoder or decoder) may compromise the performance of the other. Increasing the latent dimension benefits the encoder but may reduce the decoder's effectiveness in generating meaningful reconstruction errors. Conversely, decreasing the latent dimension enhances the decoder's ability to produce larger reconstruction errors for OOD data but may impair the encoder's discriminative capacity.

**Implications for VAE Design**    When designing a single VAE for OOD detection, it's essential to consider this trade-off:

- **For the Encoder**: Aim for a higher latent dimension to improve the separation between ID and OOD data in the latent space.

- **For the Decoder**: Consider a lower latent dimension to increase reconstruction errors for OOD data, enhancing detection based on reconstruction discrepancies.

However, finding an optimal latent dimension that satisfies both requirements within a single VAE can be challenging. Adjusting the latent dimension to favor one aspect inherently affects the other, leading to suboptimal performance in at least one component.

**Two VAEs Face No Such Trade-off**    To overcome this trade-off inherent in a single VAE, we propose using two VAEs with different latent dimensions, as discussed in the next section. By pairing a high-dimensional VAE with a low-dimensional one, we can leverage the strengths of both models without being constrained by the conflicting requirements of a single latent dimension.

D.1.2    PAIRING VAES: LEVERAGING DUAL LATENT DIMENSIONS

**Two VAEs Overcome the Trade-off**    To resolve the trade-off in latent dimension selection, we propose training two VAEs with different latent dimensions:

1. **High-Dimensional VAE**: This VAE has an overparameterized (large) latent dimension. Its encoder $q_\phi$ is capable of capturing complex features and provides informative statistics such as $v(\mathbf{x})$ and $w(\mathbf{x})$ that help discriminate between ID and OOD data.

2. **Low-Dimensional VAE**: This VAE has an underparameterized (small) latent dimension. Its decoder $p_\theta$ is constrained, leading to higher reconstruction errors $u(\mathbf{x})$ for OOD data due to its limited capacity to represent unfamiliar inputs.

By combining the strengths of both VAEs, we can effectively detect OOD data. The high-dimensional VAE's encoder excels at distinguishing ID from OOD data in the latent space, while the low-dimensional VAE's decoder amplifies reconstruction errors for OOD samples.

**Implementation Details**    In practice, we extract the following statistics:

- **From the High-Dimensional VAE**:
$$v(\mathbf{x}) = \|\boldsymbol{\mu}_{\mathbf{z}}(\mathbf{x})\|_2, \tag{26}$$
$$w(\mathbf{x}) = \|\boldsymbol{\sigma}_{\mathbf{z}}(\mathbf{x})\|_2, \tag{27}$$
where $\boldsymbol{\mu}_{\mathbf{z}}(\mathbf{x})$ and $\boldsymbol{\sigma}_{\mathbf{z}}(\mathbf{x})$ are the encoder's mean and standard deviation in the latent space.

- **From the Low-Dimensional VAE**:

$$u(\mathbf{x}) = \|\mathbf{x} - \widehat{\mathbf{x}}\|_2, \tag{28}$$

$$s(\mathbf{x}) = \|\boldsymbol{\sigma}_{\mathbf{x}}(\boldsymbol{\mu}_{\mathbf{z}}(\mathbf{x}))\|_2, \tag{29}$$

where $\widehat{\mathbf{x}}$ is the reconstructed input, and $\boldsymbol{\sigma}_{\mathbf{x}}(\boldsymbol{\mu}_{\mathbf{z}}(\mathbf{x}))$ is the decoder's standard deviation.

By integrating these statistics, we create a comprehensive feature set for OOD detection that leverages both the encoder's discriminative ability and the decoder's reconstruction error signal.

**Empirical Results** This approach has led to improvements in challenging OOD detection scenarios. For instance, when training on CIFAR-10 as the in-distribution dataset and using CIFAR-100, vertically flipped (VFlip), and horizontally flipped (HFlip) images as OOD datasets, our method achieved state-of-the-art results.

Remarkably, this was accomplished even though both VAEs, when considered individually, might have limitations:

- The **Overparameterized VAE** (high latent dimension) may overfit the training data, potentially reducing its generalization to unseen data.
- The **Underparameterized VAE** (low latent dimension) may struggle to reconstruct even some ID data accurately due to its limited capacity.

However, by combining their complementary strengths, we surpassed the performance of larger model architectures specifically designed for image data (see Table 1).

Pairing two VAEs with different latent dimensions allows us to capitalize on the advantages of both high and low-dimensional latent spaces without being constrained by the trade-offs inherent in a single model. This strategy provides a practical and effective solution for improving OOD detection performance, demonstrating that sometimes "it takes two to transcend."

### D.1.3 CONSTANT DECODER COVARIANCE

In typical VAE learning, the decoder's variance is fixed Dai et al., so it cannot be used as an inferential parameter. We initially treated the decoder as an isotropic Gaussian with a learnable scalar covariance matrix $\sigma_{\mathbf{x}}(\mathbf{z})^2 I$, where $I$ is the identity matrix and $\sigma_{\mathbf{x}}(\mathbf{z})^2$ is a learnable scalar. We later observed that the scalar $\sigma_{\mathbf{x}}(\mathbf{z})$ always converge to a small value and remains fixed for any ID or OOD data. And given that in typical VAE learning, the decoder's variance is fixed Dai et al.. We decided to use a fixed scalar as well and exclude this term from our algorithm.

This reduces the minimal sufficient statistics for encoder and decoder pair:

$$(\mu_{\mathbf{z}}(\mathbf{x}), \sigma_{\mathbf{z}}(\mathbf{x}), \mu_{\mathbf{x}}(\mathbf{z}), \sigma_{\mathbf{x}}(\mathbf{z})) \longrightarrow (\mu_{\mathbf{z}}(\mathbf{x}), \sigma_{\mathbf{z}}(\mathbf{x}), \mu_{\mathbf{x}}(\mathbf{z})) \tag{30}$$

### D.1.4 TRAINING OBJECTIVE MODIFICATION FOR STRONGER CONCENTRATION

Inspired by the well known concentration of Gaussian probability measures, to encourage stronger concentration of the latent code around the spherical shell with radius $\sqrt{m}$ for better OOD detection, we propose the following modifications to standard VAEs' loss functions:

We replace the initial KL divergence by:

$$\mathcal{D}^{\text{typical}}[Q_\phi(\mathbf{z} \mid \mu_{\mathbf{z}}(\mathbf{x}), \sigma(\mathbf{x}))\|P(\mathbf{z})] \tag{31}$$

$$=\mathcal{D}^{\text{typical}}[\mathcal{N}(\mu_{\mathbf{z}}(\mathbf{x}), \sigma_{\mathbf{z}}(\mathbf{x}))\|\mathcal{N}(0, I)] \tag{32}$$

$$=\frac{1}{2}\left(\text{tr}(\sigma_{\mathbf{z}}(\mathbf{x})) + |(\mu_{\mathbf{z}}(\mathbf{x}))^\top (\mu_{\mathbf{z}}(\mathbf{x})) - m| - m - \log\det(\sigma_{\mathbf{z}}(\mathbf{x}))\right) \tag{33}$$

where $m$ is the latent dimension.

In training, we also use Maximum Mean Discrepancy (MMD) Gretton et al. (2012) as a discriminator since we are not dealing with complex distribution but Gaussian. The MMD is computed with Gaussian kernel. This extra modification is because the above magnitude regularization does not take distribution in to account.

The final objective:

$$\mathbb{E}_{\mathbf{x}\sim P_{\text{ID}}}\mathbb{E}_{\mathbf{z}\sim Q_\phi}\mathbb{E}_{\mathbf{n}\sim\mathcal{N}}[\log P_\theta(\mathbf{x}\mid\mathbf{z})] - \mathcal{D}^{\text{typical}}[Q_\phi(\mathbf{z}\mid\mu_{\mathbf{z}}(\mathbf{x}),\sigma(\mathbf{x}))\|P(\mathbf{z}) - \text{MMD}(\mathbf{n},\mu_{\mathbf{z}}(\mathbf{x})) \tag{34}$$

The idea is that for $P_{\text{ID}}$, we encourage the latent codes to concentrate around the prior's *typical sets*. That way, $P_{\text{OOD}}$ may deviate further from $P_{\text{ID}}$ in a controllable manner. In experiments, we tried the combinations of the metric regularizer, $\mathcal{D}^{\text{typical}}$, and the distribution regularizer, MMD. This leads to two other objectives:

$$\mathbb{E}_{\mathbf{x}\sim P_{\text{ID}}}\mathbb{E}_{\mathbf{z}\sim Q_\phi}[\log P_\theta(\mathbf{x}\mid\mathbf{z})] - \mathcal{D}^{\text{typical}}[Q_\phi(\mathbf{z}\mid\mu_{\mathbf{z}}(\mathbf{x}),\sigma(\mathbf{x}))\|P(\mathbf{z})] \tag{35}$$

$$\mathbb{E}_{\mathbf{x}\sim P_{\text{ID}}}\mathbb{E}_{\mathbf{z}\sim Q_\phi}\mathbb{E}_{\mathbf{n}\sim\mathcal{N}}[\log P_\theta(\mathbf{x}\mid\mathbf{z})] - \mathcal{D}[Q_\phi(\mathbf{z}\mid\mu_{\mathbf{z}}(\mathbf{x}),\sigma(\mathbf{x}))\|P(\mathbf{z})] - \text{MMD}(\mathbf{n},\mu_{\mathbf{z}}(\mathbf{x})) \tag{36}$$

where $\mathcal{D}$ is the standard KL divergence.

But we **did not** observe a significant difference in the final AUROC different variations. We still include those attempted modifications for future work.

### D.2 FEATURE PROCESSING TO BOOST COPOD PERFORMANCES

Like most statistical algorithms, COPOD/MD is not scale invariant, and may prefer more dependency structures closer to the linear ones. When we plot the distributions of $u(\mathbf{x})$ and $v(\mathbf{x})$, we find that they exhibit extreme skewness. To make COPOD's statistical estimation easier, we process them by quantile transform. That is, for ID data, we map the the tuple of statistics' marginal distributions to $\mathcal{N}(0,1)$. To ease the low dimensional empirical copula, we also de-correlate the joint distribution of $(u(\mathbf{x}),v(\mathbf{x})),w(\mathbf{x}))$. We do so using Kessy et al. (2018)'s de-correlation method, similar to Morningstar et al. (2021).

### D.3 WIDTH AND HEIGHT OF A VECTOR INSTEAD OF ITS $l^2$ NORM TO EXTRACT COMPLEMENTARY INFORMATION

In our visual inspection, we find that the distribution of the scalar components of $(u(\mathbf{x}),v(\mathbf{x}),w(\mathbf{x}))$ can be rather uneven. For example, the visible space reconstruction $\mathbf{x}-\widehat{\mathbf{x}}$ error can be mostly low for many pixels, but very high at certain locations. These information can be washed away by the $l^2$ norm. Instead, we propose to track both $l^p$ norm and $l^q$ norm for small $p$ and large $q$.

**For small $p$, $l^p$ measures the width of a vector, while $l^q$ measures the height of a vector for big $q$.** To get a sense of how they capture complementary information, we can borrow intuition from $l^p \approx l^0$, for small $p$ and $l^q \approx l^\infty$, for large $q$. $\|\mathbf{x}\|_0$ counts the number of nonzero entries, while $\|\mathbf{x}\|_\infty$ measures the height of $\mathbf{x}$. For $\mathbf{x}$ with continuous values, however, $l^0$ norm is not useful because it always returns the dimension of $\mathbf{x}$, while $l^\infty$ norm just measures the maximum component.

**Extreme measures help screen extreme data.** We therefore use $l^p$ norm and $l^q$ norm as a continuous relaxation to capture this idea: $l^p$ norm will "count" the number of components in $\mathbf{x}$ that are unusually small, and $l^q$ norm "measures" the average height of the few biggest components. These can be more discriminitive against OOD than $l^2$ norm alone, due to the extreme (proxy for OOD) conditions they measure. We observe some minor improvements, detailed in Table 2's ablation study.

| ID: CIFAR10 | OOD | | | |
|---|---|---|---|---|
| OOD Dataset | SVHN | CIFAR100 | Hflip | Vflip |
| $l^2$ norm | 0.96 | 0.60 | **0.53** | **0.61** |
| $(l^p, l^q)$ | **0.99** | **0.62** | 0.53 | 0.61 |

Table 2: Comparing the AUC of $l^2$ norm versus our $(l^p, l^q)$ measures.

## E ABLATION STUDIES

### E.1 INDIVIDUAL STATISTICS

To empirically validate how $(u(\mathbf{x}),v(\mathbf{x}),w(\mathbf{x}))$ complement each other suggested by Theorem **??**, we use individual component alone in first stage and fit the second stage COPOD as usual. We notice

| Statistic | OOD Dataset | | | |
|---|---|---|---|---|
| | SVHN | CIFAR100 | Hflip | Vflip |
| $u(\mathbf{x})$ | 0.96 | 0.59 | 0.54 | 0.59 |
| $v(\mathbf{x})$ | 0.94 | 0.56 | 0.54 | 0.59 |
| $w(\mathbf{x})$ | 0.93 | 0.58 | 0.54 | 0.61 |
| $v(\mathbf{x})$ & $w(\mathbf{x})$ | 0.94 | 0.58 | 0.54 | 0.60 |
| $u(\mathbf{x})$ & $v(\mathbf{x})$ | 0.97 | 0.61 | 0.53 | 0.61 |
| $u(\mathbf{x})$ & $w(\mathbf{x})$ | 0.98 | 0.61 | 0.54 | 0.61 |

Table 3: COPOD on individual statistics. ID dataset is CIFAR10.

signigicant drops in performances. We fit COPOD on individual statistics $u(\mathbf{x})$, $v(\mathbf{x})$, $w(\mathbf{x})$ and show the results in Table 3. We can see that our original combination in Table 1 is better overall.

## E.2 MD

To test the efficacy of $(u(\mathbf{x}), v(\mathbf{x}), w(\mathbf{x}))$ without COPOD, we replace COPOD by a popular algorithm in OOD detection, the MD algorithm Lee et al. (2018) and report such scores in Table 1. The scores are comparable to COPOD, suggesting $(u(\mathbf{x}), v(\mathbf{x}), w(\mathbf{x}))$ is the primary contributor to our performances.

## E.3 LATENT DIMENSIONS

One hypothesis on the relationship between latent code dimension and OOD detection performance is that lowering dimension incentivizes high level semantics learning, and higher level feature learning can help discriminate OOD v.s. ID. We conducted experiments on the below latent dimensions and report their AUC based on $v(\mathbf{x})$ (norm of the latent code) in Table 4

| Latent dimension | 1 | 2 | 5 | 10 | 100 | 1000 | 3096 | 5000 |
|---|---|---|---|---|---|---|---|---|
| $v(\mathbf{x})$ AUC | 0.39 | 0.63 | 0.52 | 0.45 | 0.22 | 0.65 | 0.76 | 0.59 |

Table 4: Lower latent code dimension doesn't help to discriminate in practice.

Clearly, lowering the dimension isn't sufficient to increase OOD performances.

