# OpenReview forum: "Inference, Fast and Slow: Reinterpreting VAEs for OOD Detection"
_NeurIPS.cc/2024/Workshop/SafeGenAi — SafeGenAi Poster_

### Official Review · Reviewer_TZgk · 2024-10-08
**Good theoretical insights and formulation of using VAEs for OOD**

**Rating:** 7
**Confidence:** 3

**Review:**

This paper introduces a simple reformulation of utilizing Gaussian VAE statistics for out-of-distribution detection. The paper was well-written and contained detailed theoretical insights with a clear motivation. Additionally, compared to existing approaches, the results obtained good performances using smaller VAE models.

While the method demonstrates effectiveness in OOD detection, its generalizability to stronger OOD scenarios would also look interesting. However, for a workshop, the paper is good.

Some rewriting and adjustments can be done.

---

### Official Review · Reviewer_Zm3y · 2024-10-10
**An Interesting Approach to View VAE**

**Rating:** 5
**Confidence:** 4

**Review:**

Strength:
This paper presents an interesting approach to view VAE using fast and slow weights by the proposed Likelihood Path (LPath) Principle.

Weakness:
1) The topic is not aligned well with the theme of this workshop.
2) While the paper provides empirical evidence for the LPath Principle, there is limited theoretical grounding for why the chosen statistics are optimal for out-of-distribution (OOD) detection across various settings.
3) The selection of sufficient statistics and the application of the LPath Principle involves heuristic decisions, which may not generalize well across different datasets or other types of deep generative models (DGMs).